



# 1  Estimation of surface ammonia concentrations and emissions in

# 2  China from the polar-orbiting Infrared Atmospheric Sounding

# 3  Interferometer and  the  FY-4A  Geostationary Interferometric

# 4  Infrared  Sounder

Pu Liu[1], Jia Ding[1], Lei Liu[1], Wen Xu[2], Xuejun Liu [2]
[1] School of Earth and Environmental Sciences, Lanzhou University, Lanzhou, 730000, China
[2] College of Resources and Environmental Sciences, Centre for Resources, Environment and Food Security, Key Lab of  Plant-
Soil Interactions of MOE, China Agricultural University, Beijing 100193, China
Correspondence to: Lei Liu (liuleigeo@lzu.edu.cn)
**Abstract**
Ammonia ($NH_3$) is the most important alkaline gas in the atmosphere, which has negative effects on
biodiversity, ecosystems, soil acidification and human health. China has largest $NH_3$ emissions in the
world mainly associated with agricultural sources including nitrogen fertilizer and livestock. However,
there is still a limited number of ground monitoring sites in China, hindering our understanding of both
surface $NH_3$ concentrations and emissions.  In this study, using the polar-orbiting satellite (IASI) and
Fengyun-4 geostationary satellite (GIIRS), we analyzed the changes of hourly $NH_3$ concentrations, and
estimated surface $NH_3$ concentrations and $NH_3$ emissions in China. GIIRS-derived $NH_3$ concentration in
daytime was generally higher than that at night, with high values during 8:00-18:00. Satellite-derived
surface $NH_3$ concentration was generally consistent with the ground observation data with R-square at





0.72-0.81 and slope equal to 1.03. Satellite-based $NH_3$ emissions ranged from 12.99-17.77 Tg N yr$^{-1}$
during 2008-2019. Spatially, high values of $NH_3$ emissions mainly occurred in the North China Plain,
Northeast China and Sichuan Basin, while low values were mainly distributed in western China (Qinghai-
Tibet Plateau). Our study shows a high predictive power of using satellite data to estimate surface $NH_3$
concentration and $NH_3$ emissions over multiple temporal and spatial scales, which provide an important
reference for understanding $NH_3$ changes over China.
**1 Introduction**
Ammonia ($NH_3$) is a highly active gas in the atmosphere and the most important alkaline gas, playing an
important role in atmospheric chemistry (Fowler et al., 2013). $NH_3$ reacts with acid pollutants ($SO_2$ and
NOx) to form fine particulate matters (such as PM2.5), leading to haze pollution. In addition, the
deposition of $NH_3$ and $NH4^+$ could also cause environmental problems such as water eutrophication,
biodiversity loss and soil acidification (Paerl et al., 2014). To provide a scientific basis of dealing with
$NH_3$ pollution, it is urgent to accurately estimate both surface $NH_3$ concentrations and emissions in China.

Surface $NH_3$ concentration can be estimated by ground measurements and model simulations. Ground
measurements are considered to be the most accurate quantitative method. Current national $NH_3$
observation networks in China include the National Nitrogen Deposition Monitoring Network (NNDMN)
established by China Agricultural University (Xu et al., 2015) and the Ammonia Monitoring Network
(AMoN-China) established based on the Chinese Ecosystem Research Network (CERN) (Pan et al., 2018).
The NNDMN can measure ground $NH_3$ concentrations since 2010, while the AMoN-China only made



the measurements in 2015-2016. The above two monitoring networks both monitor surface $NH_3$
concentration on a monthly basis, and lack monitoring of the hourly $NH_3$ changes. Some studies have
carried out research on the intra-day/hourly changes of $NH_3$ concentrations based on ground observations.
Werner et al. (2017) measured hourly $NH_3$ concentration in 2012 at the Harwell site in the United
Kingdom, and found that high $NH_3$ concentration usually occurred in the afternoon. Similarly, Kutzner
et al. (2021) observed the hourly $NH_3$ concentration on the SIRTA Observatory in Paris and found $NH_3$
concentration is highest in the late afternoon. Pandolfi et al. (2012) studied the day-night cycle of $NH_3$
concentration at the two stations in Barcelona in summer, and found that the $NH_3$ concentration was
highly associated with local meteorology and traffic emissions. However, there are still a limited number
of monitoring sites on the hourly $NH_3$ changes in China.

Agricultural fertilizer and livestock production have led to a large amount of $NH_3$ emissions. China's
cultivated land area accounts for only 8% of the world, but it consumes about 30% of the world's nitrogen
(N) fertilizer. Estimation of $NH_3$ emissions is mainly based on a bottom-up method, using $NH_3$ source
statistics (fertilization, animal husbandry, etc.) and emission factors. Zhou et al. calculated the annual
farmland $NH_3$ emission ($3.96 \pm 0.76$ Tg N $yr^{-1}$) in China based on the bottom-up method, which is 40%
higher than the emission in IPCC tier 1 guidelines (Zhou et al., 2016). Zhang et al. reassessed China's
$NH_3$ emissions based on the mass balance method, and found $NH_3$ emissions increased from $12.1 \pm 0.8$ in
2000 to $15.6 \pm 0.9$ Tg N $yr^{-1}$ in 2015, with an annual growth rate of 1.9% (Liu et al., 2017). Fu et al.
estimated that China's $NH_3$ emissions increased from 4.7 in 1980 to 11 Tg N $yr^{-1}$ in 2016. Although many
studies regarding $NH_3$ emission have been carried out in China, great uncertainties and large ranges (7-





16 Tg N yr$^{-1}$) still existed in the estimates of China's NH$_3$ emissions (Dong et al., 2010; Huang et al.,
2012; Kang et al., 2016).

Besides the bottom-up estimates, some studies used data assimilation methods by ground monitoring data
to constrain NH$_3$ emission estimates. Paulot et al. (2014) assimilated the GEOS-Chem with ground
observations of wet N$_r$ deposition, and estimated China's NH$_3$ emission as 8.4 Tg N yr$^{-1}$ in 2008 with
seasonal NH$_3$ emission peaked in summer. Gilliland et al. (2003) used the data assimilation method by
the air quality model (CMAQ) with the wet NH$_4^+$ concentration data from the USA National Atmospheric
Deposition Program Network, and found that obvious seasonal differences appeared in NH$_3$ emissions
linked to N fertilizer and temperature. Kong et al. (2019) carried out inversion through assimilating
surface AMoN NH$_3$ observation data, and improved the accuracy of temporal and spatial pattern of NH$_3$
emission in China.

In recent years, atmospheric remote sensing has developed rapidly, which can monitor NH$_3$ at a global
scale including the polar-orbiting satellite instruments such as the Tropospheric emission spectrometer
(TES), Infrared Atmospheric Sounding Interferometer (IASI), Cross-track Infrared Sounder (CrIS),
Atmospheric Infrared Sounder (AIRS), and Greenhouse Gases Observing Satellite (GOSAT) (Someya et
al., 2020). There have been many studies reporting the effectiveness of using satellite data to study NH$_3$
dynamics. Pinder et al. (2011) found that TES observation can capture spatial-temporal NH$_3$ patterns
compared with surface measurement; Van Damme et al. (2014b) studied the seasonal and annual NH$_3$
changes in the northern and southern hemisphere using the IASI NH$_3$ column data, and found that the





seasonality in the southern hemisphere is mainly related to biomass burning; Shephard and Cady-Pereira
(2015) developed the CrIS NH$_3$ inversion algorithms, and found that CrIS can capture the global spatial
distribution of NH$_3$ concentration; Warner et al. (2016) identified the main hotspots of agricultural NH$_3$
regions using AIRS, such as South Asia, China, the United States and some parts of Europe, and found
that NH$_3$ concentrations increased at these agricultural regions since 2003. Besides, some studies also
used the satellite measurements to improve the estimates of NH$_3$ emissions. Zhang et al. (2017) developed
a top-down inversion method by using TES NH$_3$ observation to quantify China's NH$_3$ emission, and
obtained annual NH$_3$ emission as 11.7 Tg N yr$^{-1}$ in 2008. Marais et al. (2021) estimated NH$_3$ emissions
in the UK based on IASI and CrIS and found the relative errors of IASI-derived NH$_3$ emissions were 11-
36% and 9-27% respectively; Van Damme et al. (2018) used the high-resolution IASI NH$_3$ maps to
identify, classify and quantify NH$_3$ emission hotspots in the world, which is helpful to understand the
man-made point NH$_3$ sources; Dammers et al. (2019) identified global 249 NH$_3$ emission point sources
based on CrIS, which is about 2.5 times higher than that reported in the HTAPv2 emissions.

Besides the polar-orbiting satellite, China's Geostationary Interferometric Infrared Sounder (GIIRS)
onboard the Chinese FY-4A satellite can measure hourly changes of atmospheric NH$_3$ almost all of Asia
per day, which provide great potential to study the diel cycle of NH$_3$. In this study, GIIRS was used to
study the NH$_3$ diel cycle (hourly changes), which is essential for understanding the differences between
different times in a day by the polar-orbiting satellites (such as IASI at 9:30 and CrIS at 13:30). Second,
the surface NH$_3$ concentration in China is estimated based on both GIIRS and IASI, which was then
compared with the NNDMN. Third, NH$_3$ emission in China are calculated based on satellite-derived





surface $NH_3$ concentration and the feedback relationship between surface $NH_3$ concentration and emission
by a chemistry transport model (GEOS-Chem). Finally, the spatial-temporal characteristics of satellite-
derived surface $NH_3$ concentration and emission were analyzed, and the uncertainties were discussed.

## 107 2 Data and methods

### 108 2.1 Satellite GIIRS $NH_3$

The Geostationary Interferometric Infrared Sounder (GIIRS) onboard the Fengyun-4A geostationary
satellite (FY-4A) launched by China in 2016 is the world's first hyperspectral atmospheric infrared
sounder. The FY-4A GIIRS detected spectral range is 700-2250 $cm^{-1}$, including 1648 spectral channels
and 14 radiations imaging channels, covering visible light, short, medium and long wave infrared bands.
The spatial resolution of the detector is 2.0 km in the visible band and 16 km in the infrared bands. It
covers almost the whole of Asia and scans 10 times a day. The GIIRS can detect the temperature and
humidity profiles and trace gases at high frequency.

$NH_3$ has two large absorption characteristics in the long wave infrared (about 930 $cm^{-1}$ and 965 $cm^{-1}$).
The contribution of $NH_3$ to the brightness temperature of these two bands is between 2-4 k. The core of
inversion algorithms is based on the so-called hyperspectral radiation index (HRI), which quantifies the
spectral characteristics of $NH_3$. The HRI index depends on whether the satellite instrument detects the
presence of $NH_3$. The average value of HRI is 0 with the standard deviation as 1, and the HRI range is [-
1,1]. The algorithms for estimating IASI $NH_3$ column concentration is to convert HRI into a column using
the so-called neural network (Clarisse et al., 2021).





In this study, we used hourly $NH_3$ concentrations during 2019-2020 (from November in 2019 to October
in 2020) to study $NH_3$ diel cycle with a resolution of 0.5°. The original data is in H5 format, and the unit
of $NH_3$ is molec.$cm^{-2}$. The data is processed with MATLAB software. First, observations with
considerable uncertainties (relative error exceeds 50%) and high clouds (cloud cover exceeds 20%) were
removed. Secondly, the world standard time (UTC) by GIIRS is converted to local time, and the data in
H5 format is converted to TIFF data.
**2.2 Satellite IASI $NH_3$**
IASI instrument is on board the polar solar synchronous Metop-A platform. It has been running stably
since 2006 to measure the infrared radiation emitted by the earth (Van Damme et al., 2014a). IASI can
measure the infrared radiation emitted by the earth's surface and atmosphere in the spectral range of 645-
2760 $cm^{-1}$. It can observe the world twice a day, and cross the equator at 9:30 and 21:30 local time, with
a spatial resolution of 12 km at nadir. However, only daytime satellite measurements are used, because
nighttime measurements usually have greater uncertainties related to thermal contrasts (Van Damme et
al., 2017).

The cloud free reanalysis product of total $NH_3$ column (v3R-ERA5) was used here. The properties of
IASI $NH_3$ data include $NH_3$ column concentration, longitude, latitude, measurement time, cloud cover,
uncertainty, solar zenith angle and other parameters. The daily $NH_3$ column concentration from 2008 to
2019 was used. The format of the original data is NC format, and the unit is molec.$cm^{-2}$. The observation
data of with cloud cover larger than 20% and the uncertainty above 50% are removed. We gridded the
data to 0.1° by using the arithmetic average methods.



## 2.3 Ground NH₃ measurements

Surface NH$_3$ concentrations in the NNDMN were used to compare with the satellite estimates including

43 observation stations. The land types of the NNDMN sites cover cities, farmland, coastal areas, forests

and grasslands. Measurements during the period from January 2010 to December 2015 by the NNDMN

were used. Surface NH$_3$ concentrations were measured using an active DELTA (DEnuder for Long-Term

Atmospheric sampling) (Flechard et al., 2011). For the hourly measurements, we collected the data from

the published papers including 5 sites including Xianghe (39.75 °N, 116.96 °E, 2017.12 -2018.2) (He et

al., 2020), Fudan University (31.30 °N, 121.50 °E, 2013.7.1-2014.9.30) (Wang et al., 2015), Dianhushan

(31.09 °N, 120.98 °E, 2014.3.1-2014.6.30) (Wang et al., 2015), Jinshan Chemical Industry Park (30.73 °N,

121.27 °E, 2014.1.6-2014.6.30) (Wang et al., 2015), and Gucheng (39.15 °N, 115.73 °E, 2016.3-2017.5)

(Kuang et al., 2020). The Xianghe site in Hebei Province and Dianhushan site in Shanghai represent rural

environments; Jinshan Chemical Industry Park represents industrial environments; Gucheng site in Hebei

and the Fudan University site in Shanghai represent urban environments.

## 2.4 GEOS-Chem

The GEOS-Chem model version 12.3.0 is a three-dimensional chemistry transport model developed by

Harvard University, which has been widely used in the field of atmospheric studies (Eastham et al., 2014).

The nested regional model in Asia was used in this study driven by assimilated GEOS-5 meteorological

data at a horizontal resolution of $1/2° \times 2/3°$. Dry deposition calculation in GEOS-Chem follows a

standard resistance-in-series model (Wesely, 1989), while wet deposition includes both convective

updraft and large-scale precipitation scavenging (Jacob, 1999). The GEOS-Chem model here does not

consider land-atmosphere bi-directional NH$_3$ exchange and the NH$_3$ flux was parameterized as uncoupled





emission and dry deposition processes. Anthropogenic emissions over China were from the Regional
Emission in Asia (REAS-v2) inventory. The GEOS-Chem outputs of $NH_3$ concentrations include 47
layers from the ground to the top of atmosphere, which is used to capture $NH_3$ vertical profiles. The
feedback between surface $NH_3$ concentration and emissions was also calculated by GEOS-Chem.
**2.5 Satellite-based surface $NH_3$ estimates and emissions**
Surface $NH_3$ concentrations were estimated using the satellite $NH_3$ columns as well as $NH_3$ vertical
profiles. To gain the continuous vertical $NH_3$ profile, the Gaussian function was used to fit the 47 layers'
$NH_3$ concentrations. A three parameter Gaussian function was used to fit $NH_3$ vertical profiles at each
grid box from GEOS-Chem according to previous studies (Liu et al., 2019).
$\rho(Z) = \sum_{i=1}^{n} \rho_{max,i} e^{-(\frac{Z-Z_{0,i}}{\sigma_i})^2}$ , (1)
where Z is the height of a layer in the ACTM; $\rho_{max}$, Zo and σ are the maximum of $NH_3$ concentration,
the corresponding height with the maximum of $NH_3$ concentration and the thickness of $NH_3$ concentration
layer (one standard error of Gaussian function).

The satellite-derived $NH_3$ concentration at the height of $h_G$ can be calculated as:
$S_{G\_NH_3} = S_{trop} \times \frac{\rho(h_G)}{\int_0^{h_{trop}} \rho(Z)dx} \times \frac{G_{ACTM}^{1-24}}{G_{ACTM}^{overpass}}$ , (2)
where $\frac{\rho(h_G)}{\int_0^{h_{trop}} \rho(Z)dx}$ represents the ratio of $NH_3$ concentration at the height of $h_G$ to total columns
($\int_0^{h_{trop}} \rho(Z) dx$); $S_{trop}$ represents satellite-derived $NH_3$ columns; $\frac{G_{ACTM}^{1-24}}{G_{ACTM}^{overpass}}$ is the ratio of average surface
$NH_3$ concentration ($G_{ACTM}^{1-24}$) to that at satellite overpass time ($G_{ACTM}^{overpass}$) by an ACTM.





The mass balance method (Geddes and Martin, 2017; Cooper et al., 2017; Lamsal et al., 2011) was used
to exploit the feedback ratio of surface NH$_3$ concentrations and NH$_3$ emissions (Marais et al., 2021):
$$E_s = S_{G\_NH_3} \times \left(\frac{E}{G_{G\_NH_3}}\right)_m,$$  (3)
where $E_s$ is satellite-based NH$_3$ emissions; $S_{G\_NH_3}$ is the satellite-derived surface NH$_3$ concentrations;
$\left(\frac{E}{G_{G\_NH_3}}\right)_m$ is the ratio of surface NH$_3$ concentrations and NH$_3$ emissions simulated by the GEOS-Chem.

## 3 Results and discussions

### 3.1 GIIRS-based hourly NH$_3$ concentrations during 2019-2020

Fig. 1 shows the hourly NH$_3$ concentrations observed from GIIRS during 2019-2020. Daytime NH$_3$
columns were significantly higher than those at night. The intra-day hourly NH$_3$ columns showed an
overall increase first, then a decrease, with high values during 8:00-18:00. The increase in temperature
enhanced the volatilization of NH$_3$, which may explain high values of NH$_3$ concentration during the
daytime.

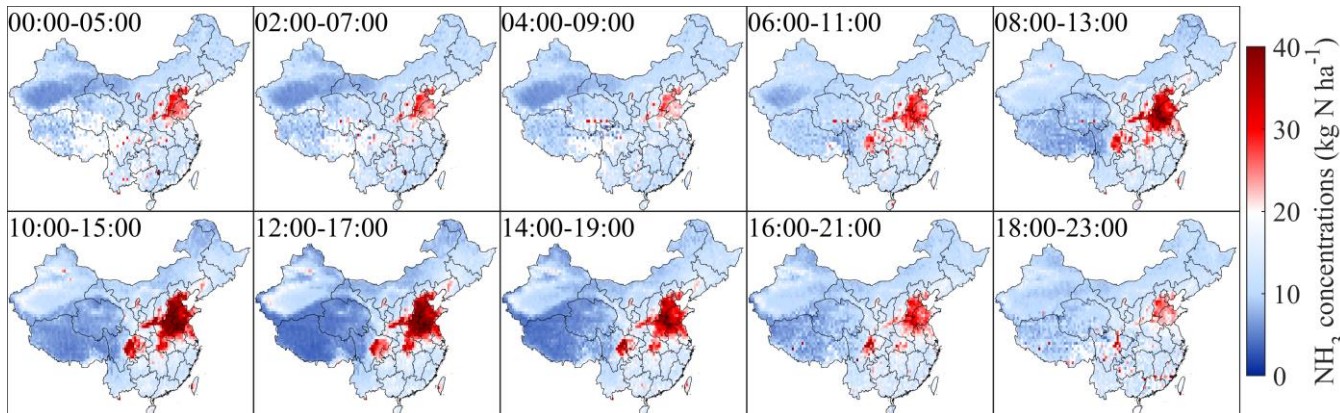

**Figure 1.** Annual NH$_3$ columns for each of the 10 GIIRS overpass time periods during 2019-2020



Ground-based measurements of hourly $NH_3$ concentration are very lacking, and the timespan may be
different from GIIRS measurements. Here we only used them to show the hourly patterns of $NH_3$
concentration (Fig. 2). The Xianghe site in Hebei Province and Dianhushan site in Shanghai represent
rural environments; Jinshan Chemical Industry Park represents industrial environments; Gucheng site in
Hebei and the Fudan University site in Shanghai represent urban environments.

$NH_3$ concentration in the rural environment basically shows a normal distribution, and high $NH_3$
concentration generally appears between 8:00-18:00, which may be related to agricultural activities and
temperature. In the industrial environment (Jinshan Chemical Industry Park, JSP), $NH_3$ concentration
fluctuates irregularly, and two peaks appear at 12:00 and 6:00-8:00, while for other time $NH_3$
concentration tends to be stable. In the urban environment, the changes of $NH_3$ concentration by satellite
at the Gucheng site are more consistent with ground monitoring, showing a clear peak around 9:00-13:00;
$NH_3$ concentration at the Fudan University site gradually decreases from the morning peak to the
afternoon. The evaporation of dew may drive the $NH_3$ increase from the morning to the noon (Wang et
al., 2015). $NH_3$ concentration in cities (Gucheng and Fudan University) has double peaks between 6:00-
8:00 and 18:00-19:00, which may be also related with traffic emissions. In summary, except the industrial
sites, hourly $NH_3$ in China have a large variability between day and night and the hourly $NH_3$ patterns
affected by many factors, of which anthropogenic emissions and temperature seem to be the most
important driving factors.



**Figure 2.** GIIRS-based and measured hourly NH$_3$ concentrations at five sites including Jinshan Chemical Industry Park (JSP, a), Xianghe (XH, b), Dianhushan (DSL, c), Fudan University (FDU, d) and Gucheng (GC, e)





Spatial distribution of GIIRS-based surface NH₃ concentration across China had large variability. High
NH₃ concentration is mainly concentrated in the North China Plain, with an average of 12 μg N m⁻³,
followed by Northeastern China, Xinjiang and the middle and lower reaches of the Yangtze River (4-8
μg N m⁻³); low values (<2 μg N m⁻³) are mainly concentrated in the Qinghai-Tibet Plateau. High surface
NH₃ concentration appeared in July (6.79 μg N m⁻³), and the lowest values appeared in November (3.26
μg N m⁻³). There are obvious seasonal changes in surface NH₃ concentration in the NCP with high values
in summer, and low values in winter, related to both agricultural N fertilizer and higher temperature.

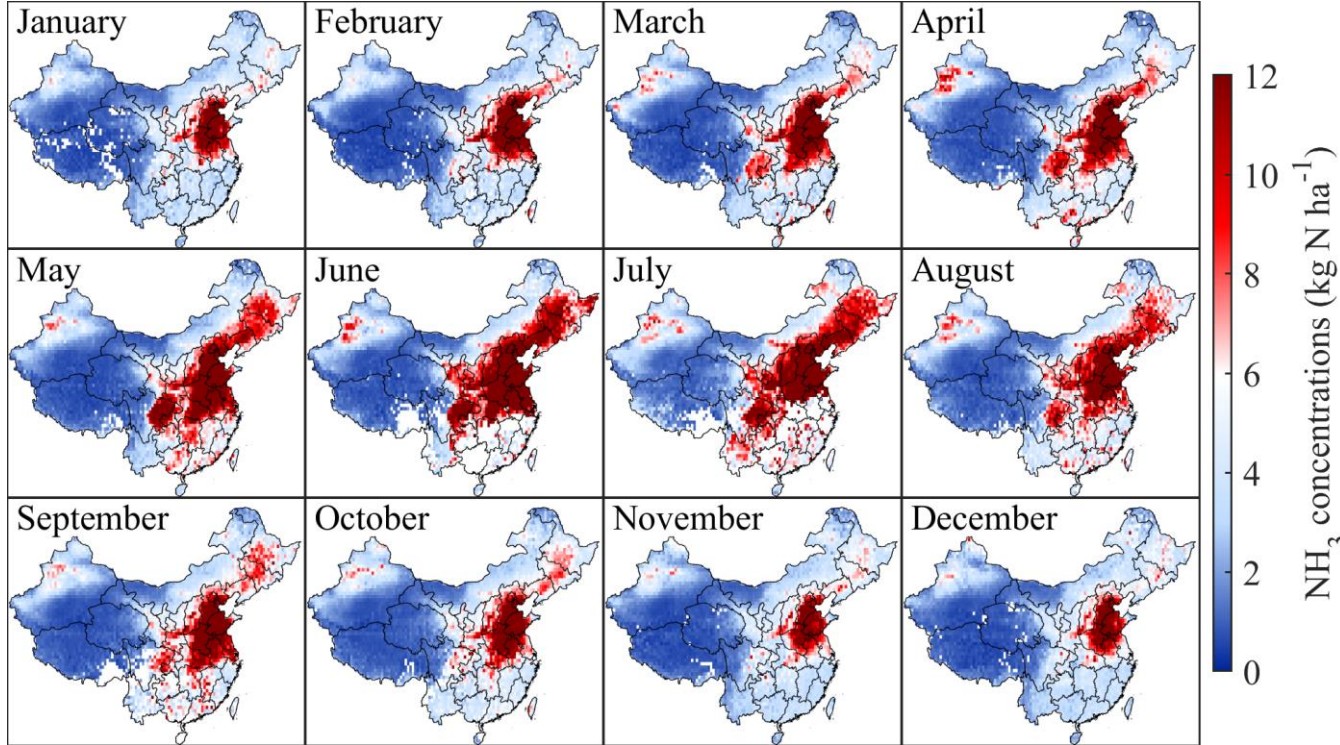


**Figure 3.** Spatial distribution of the monthly surface NH₃ concentration in China by GIIRS in 2019-2020

### 3.2 IASI-based NH₃ surface concentrations

The observation data of the NNDMN in China was collected to compare with IASI-derived surface $NH_3$

concentration. In general, a good consistency was found between measurements and satellite estimates

with the regression $R^2$ as 0.72 and the RMSE as 2.24 µg N m$^{-3}$. The coefficient of the fitted line is 1.03≈1

and the bias is 2.59%.

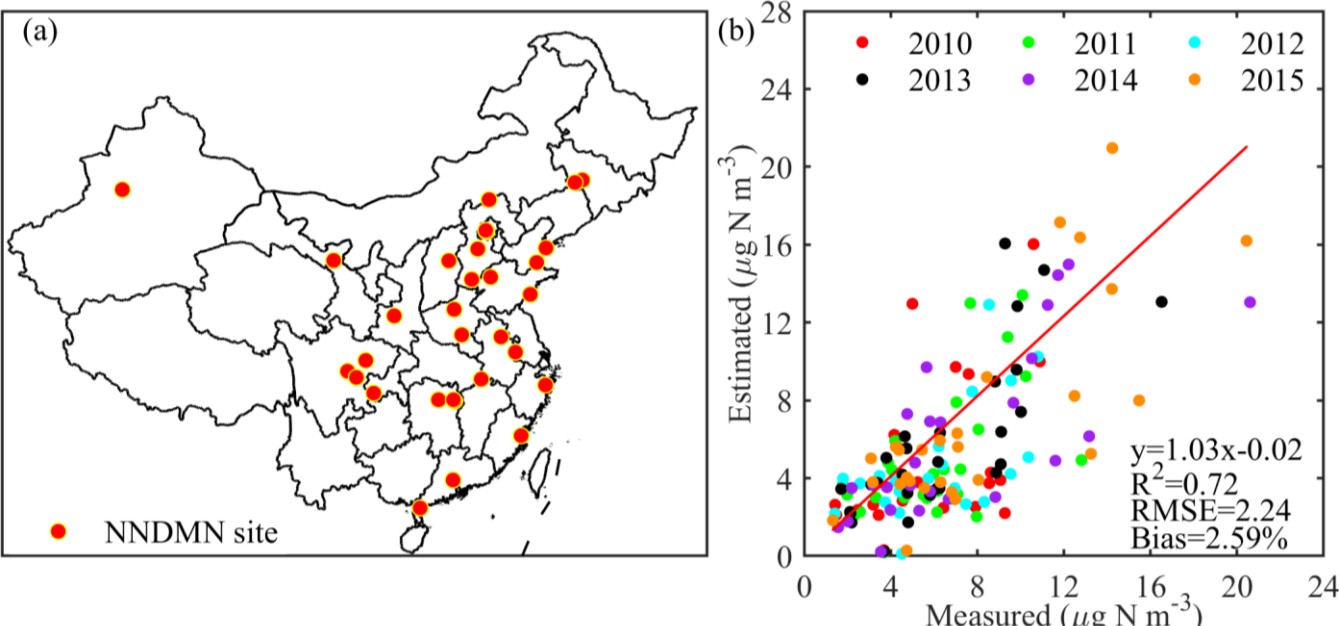

**Figure 4.** Comparison of IASI surface $NH_3$ concentrations with NNDMN measurements. (a) the locations

of NNDMN; (b) the regression results between satellite-estimates and measurements.

Monthly regression $R^2$ between the satellite-derived $NH_3$ concentration and the measured $NH_3$ was 0.37-

0.81. The regression $R^2$ reached the maximum value of 0.81 in July and August. The RMSE ranged from

2.22- 3.54 µg N m$^{-3}$, which reached the maximum value of 3.54 µg N m$^{-3}$ in July, and reached the smallest

in March (2.22 µg N m$^{-3}$). The bias is basically less than 30% for all months, and reached the minimum



value of 0.71% in February, indicating that the monthly IASI-derived surface concentration obtained are
consistent with measurements.

**Figure 5.** Comparison of monthly average values of IASI-derived and observed NH₃ surface

concentrations in 2010-2015

Fig. 6 shows the monthly changes of surface NH₃ concentrations in Huinong County in Ningxia from
2010 to 2015 for a total of 72 months during 2010-2015. Surface NH₃ concentration retrieved by IASI
was compared with the observation data at Huinong. The highest value of each year basically appeared





from June to August, and the lowest values appeared from December to January. In the past 6 years, the
maximum measured NH₃ concentrations appeared in July 2015 (18.9 μg N m⁻³), and the minimum
appeared in November 2012 (0.6 μg N m⁻³). The observation data and satellite data have the same seasonal
changes.

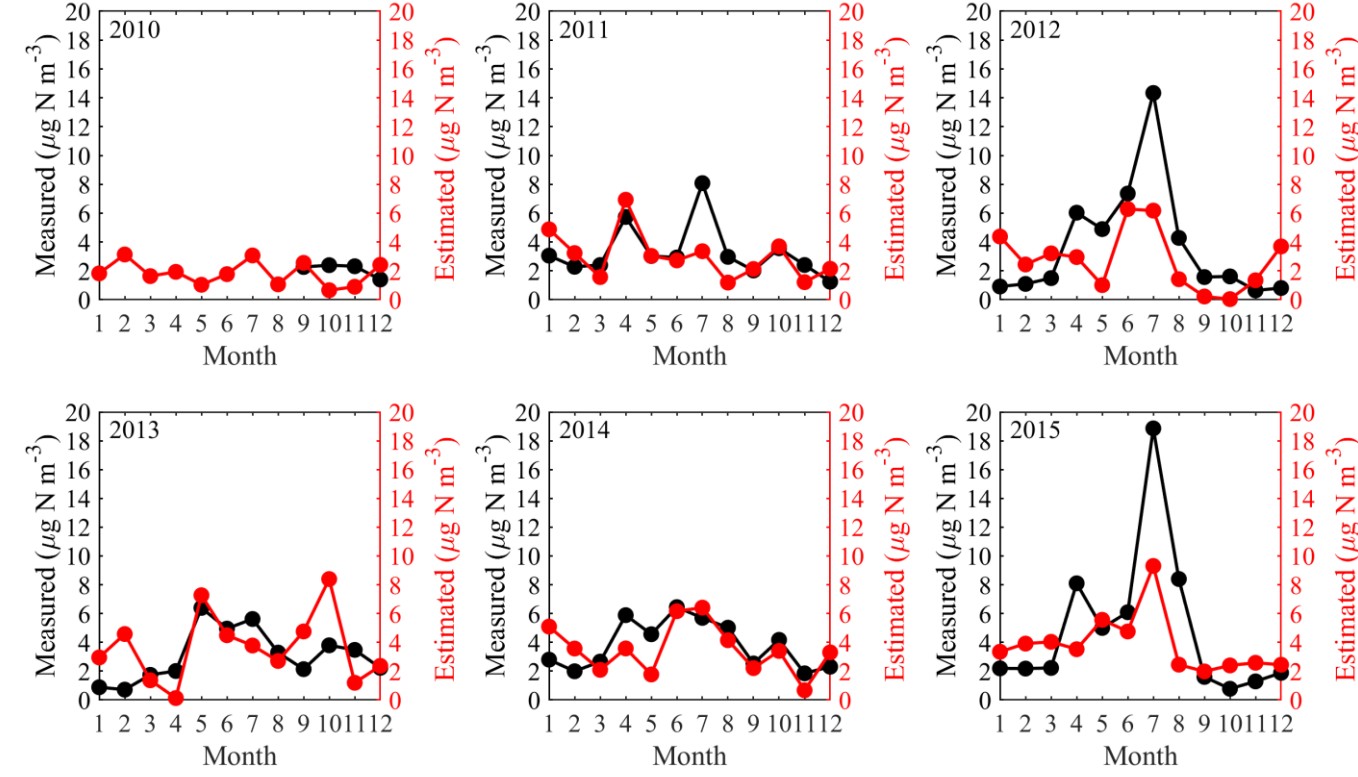


**Figure 6.** Monthly changes of NH₃ concentrations in Huinong County in Ningxia from 2010 to 2015 for
a total of 72 months during 2010-2015.

Fig. 3 and Fig. 7 show the spatial distribution of GIIRS-derived and IASI-derived surface NH₃
concentration in 2019. The spatial distribution and gradients of surface NH₃ concentration by the GIIRS
and IASI have the same gradients from eastern to western regions. One notable difference occurred in the



middle and lower reaches of the Yangtze River in June and July since the GIIRS observations were
affected by clouds and have missing data.

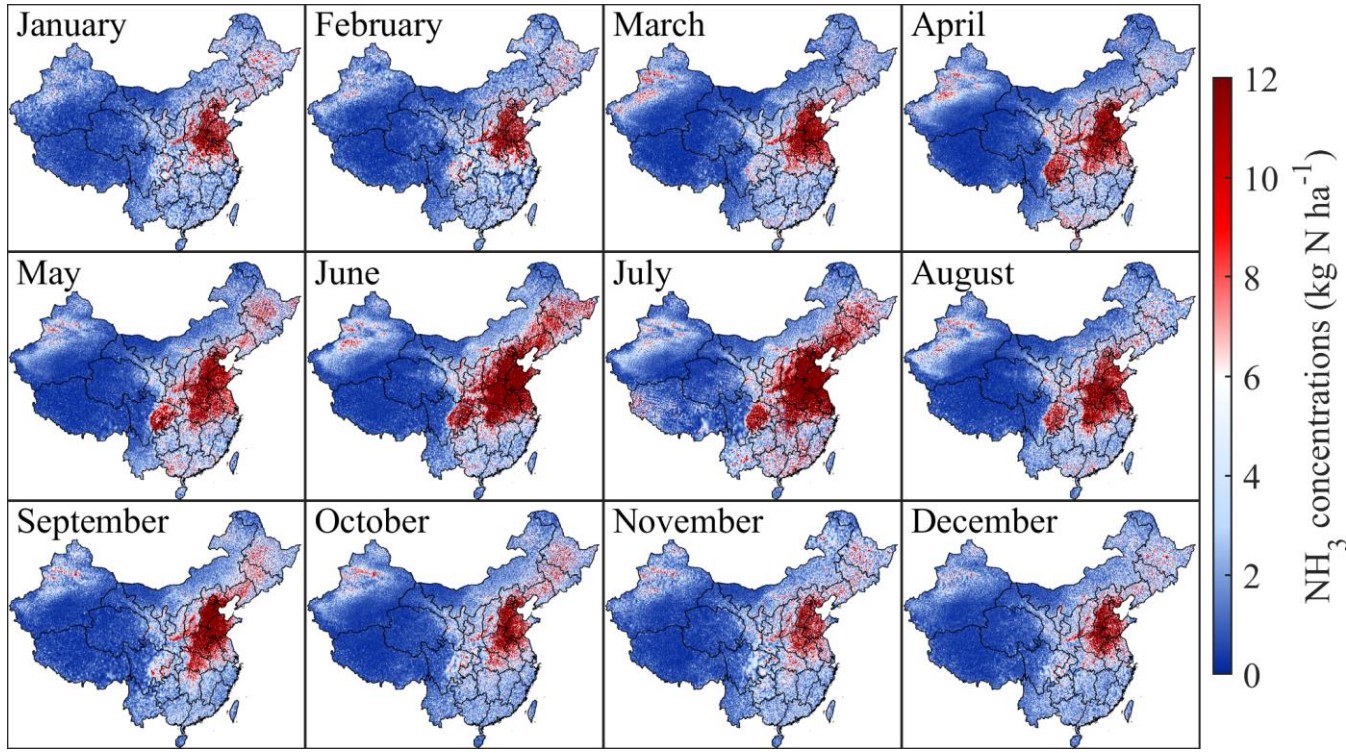

**Figure 7.** Spatial distribution of the monthly surface NH₃ concentration in China by IASI in 2019

**3.3 IASI-derived NH₃ emissions**
Based on the top-down estimates, China's NH₃ emissions ranged from 12.99-17.77 Tg N yr⁻¹ during
2008-2019. From 2008 to 2015, NH₃ emissions increased from 12.99 Tg N yr⁻¹ to 17.06 Tg N yr⁻¹. Since
2008, the temperature in China has risen steadily (Ding et al., 2007), which promotes the volatilization of
NH₃, which partly explains the increase in NH₃ emissions from 2008 to 2015. After 2015, NH₃ emissions
fluctuated and changed slightly (16.08-17.77 Tg N yr⁻¹). Compared with other studies, the change of NH₃
emissions from 2008 to 2015 is consistent with previous estimates, and the overall NH₃ emissions show



an upward trend (Zhang et al., 2021; Fu et al., 2020; Zhang et al., 2018; Ma, 2020; Kang et al., 2016).
Our estimates are on the rise as a whole, but the calculated values are generally lower than those by Fu et
al. (around 15 Tg N yr$^{-1}$), but larger than those by EDGAR and Kang et al. (2016).

In terms of spatial distribution, high $NH_3$ emissions are generally distributed in the North China Plain,
Sichuan Basin, Northeast China and Xinjiang, while the low values are mainly distributed in the
Southwest China (especially Qinghai-Tibet Plateau). The North China Plain is China's granary, with
developed agriculture and animal husbandry, high population densities and strong human activities
(including vehicle emissions). In contrast, South China is rich in rainfall, which promotes the deposition
of $NH_3$ and suppresses its volatilization to a certain extent.

The spatial distribution of $NH_3$ emissions in January, April, July and October were selected to characterize
the seasonal variations. The average emissions for the four months were 1.15 kg N ha$^{-1}$, 1.31 kg N ha$^{-1}$,
2.31 kg N ha$^{-1}$ and 1.16 kg N ha$^{-1}$ in January, April, July and October, indicating that $NH_3$ emission is the
highest in summer and the lowest in winter. The annual average emission intensity of 2019 is 16.53 kg N
ha$^{-1}$ (0.09-313.47 kg N ha$^{-1}$). Fig. 8b shows the monthly changes in $NH_3$ emissions, which basically shows
a normal distribution. High values are generally distributed in June and July, and low values are generally
distributed in November and December. It reached the maximum monthly emission of 2.6 Tg N m$^{-1}$ in
July 2019 and reached the minimum monthly emission in November and December (1.1 Tg N m$^{-1}$).





**Figure 8.** Annual changes of NH₃ emissions (a), monthly changes of NH₃ emissions in 2019 (b) and

spatial distribution of NH₃ emissions by month in 2019 (c).


## 4 Limitations and outlook

This study developed satellite-based surface NH$_3$ concentration and emissions in China based on remote sensing data (IASI and GIIRS). However, several limitations have been identified in this study. First, the Fengyun geostationary satellite used in this study can achieve hourly NH$_3$ concentrations, but the time series are still very short (2019-2020), and satellite observations are affected by clouds and meteorological conditions, resulting in missing values in parts of the Yangtze River Basin. Second, the spatial resolution of the NH$_3$ vertical profile simulated by the atmospheric model is relatively coarse (0.5 degrees). In order to make it consistent with the spatial resolution of the remote sensing data, the outputs of GEOS-Chem (vertical profiles and feedback ratio between emissions and surface NH$_3$ concentrations) were interpolated through resampling methods. Third, at present, there are more and more satellite sensors (GOSAT, CrIS, AIRS, etc.) that can monitor NH$_3$ concentration. This study only used IASI and GIIRS, and in the future, data from different satellites can be merged to analyze NH$_3$ changes on multiple temporal and spatial scales.

## 5 Conclusion

We use GIIRS to study the NH$_3$ diel cycle, and estimated surface NH$_3$ concentrations and emissions based on IASI in China. There are obvious hourly changes in NH$_3$ concentration in China using GIIRS. Overall, NH$_3$ concentration was larger at daytime than nighttime in China. Hourly NH$_3$ concentrations at different land use show different patterns, but high values generally appear in 8:00-18:00. Comparing IASI-derived and observed NH$_3$ surface concentrations by NNDMN from 2010 to 2015, the coefficient of the fitted line is 1.03≈1, and low bias is 3%, indicating satellite estimates have good consistency with the



measurements. IASI-derived China's NH$_3$ emissions ranged from 12.99-17.77 Tg N yr$^{-1}$ during 2008-
2019, among which NH$_3$ emissions increased from 2008 to 2015. The emission intensity of NH$_3$ in China
presents a strong spatial heterogeneity, showing high in the eastern and low in the western. The high
values are mainly distributed in the North China Plain and Sichuan Basin. High values are generally
distributed in summer, and low values generally occurred in winter. This study provides an important
reference basis for the formulation of NH$_3$ pollution prevention and control policy in China.

**Data availability**

IASI data were obtained from the https://iasi.aeris-data.fr/nh3/. The GIIRS data were obtained from
https://zenodo.org/record/4540024.

**Author contributions**

The study was conceived by LL, and data analysis were performed by JD. The paper was written by PL,
with editing from WX and XL. LL was involved in obtaining the project grant and supervised the study.

**Competing interests**

The contact author has declared that neither they nor their co-authors have any competing interests.

**Acknowledgements**

This study is supported by the National Natural Science Foundation of China (42001347, 41705130, and
41922037) and the Chinese State Key Research & Development Programme (2017YFC0210100,



2017YFD0200101). The analysis in this study is supported by the Supercomputing Center of Lanzhou
University.

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
