# Peer review of "China from the polar-orbiting Infrared Atmospheric Sounding"

_Atmospheric Chemistry and Physics, 2022_

## Author Comment (AC1)

**Comments:**

This paper is analyzing hourly variation of NH3 concentrations and quantifying surface NH3 concentrations and NH3 emissions in China, using observations from GIIRS and IASI. A three parameter Gaussian function is used to fit NH3 vertical profiles from GEOS-Chem and get information of NH3 concentration at different heights. Surface NH3 concentrations and total NH3 emissions are estimated based on the mass balance method and ratio from GEOS-Chem.

It was found that diurnal NH3 concentrations are larger than nightly NH3 concentrations. A good agreement is obtained between the ground measurements and the estimated. The NH3 emissions range from 12.99 to 17.77 Tg N yr-1 between 2008 and 2019 in China. The paper also discussed the uncertainties and capabilities of the method. The topics of paper fits the scope of ACP and the scientific idea is new. The article is generally well written and easy to follow. I have the following comments of the paper but I am supportive of publications if these aspects can be addressed.

We thank the reviewer for your time and helpful comments. Our point-by-point response is enclosed.

**Major concerns:**

1. Please indicate the basis for the satellite data quality screening and the number of valid pixels after eliminating invalid pixels. If the proportion of remaining valid pixels is low, the study results will be misleading and appropriate data supplementation should be performed.

We thank the reviewer for the comment. Since the quality of satellite data was greatly affected by cloud cover, we deleted all data with recorded cloud cover >20%, which was more stringent than previous studies (Wang et al., 2020). Besides, the quality of satellite data was also affected by other factors, such

as retrieval method and inversion algorithm. We also deleted the observations with the uncertainty higher than 50% (Fortems - Cheiney et al., 2016).

We performed relatively stringent screening of the observations. Further quality constraint is feasible, but more satellite observations will be missing, affecting the spatial continuity of the NH3 column and the estimated NH3 emissions. We supplemented Fig. S1 showing the effective pixel count of GIIRS and IASI with the above filtering conditions. The total number of pixels covered by GIIRS (spatial resolution  $0.5^{\circ}$ ) in China is about 3840, and the total number of pixels covered by IASI (spatial resolution of  $0.1^{\circ}$ ) is about 96170.

Fig. S1a shows the number of effective pixels for the monthly average NH3 column of GIIRS in each overpass period (defined as 2 hours interval) during 2019.11-2020.10. The red dotted line represents the total number of pixels (the sum of the records of 10 overpass periods with 2 hours interval), and the inside of the bar graph is marked as the ratio of the number of valid pixels to the total pixels in a single overpass period. All the monthly average NH3 column of GIIRS accounted for more than half of the total effective pixels, and the minimum value of the effective pixels for each overpass period exceeded 25%. GIIRS had the higher number of effective pixels in March, April, and May, while the number of effective pixels in January, July, and December were relatively low, which was consistent with the results (Fig. 3) of monthly average surface NH3 concentration by GIIRS in China. It should be noted that due to the lack of the observations at 24:00, the observations at 23:00 were not shown in the Fig. S1a (the observaed coverage is extremely limited).

Fig. S1b shows the effective pixel count of monthly average NH3 column by IASI from 2008 to 2019. The red dotted line represents the total number of pixels, and the outer labels of the pie chart represent the ratio of the effective pixels to the total pixels over China. Here, we only marked a few months with the high proportion in the pie chart. The effective pixels of IASI NH3 column showed an overall increasing trend during 2008-2019. The number of IASI NH3 effective pixels was significantly higher in summer than in winter, and the effective pixel in July 2015 accounted for the highest proportion (>50%). Although the proportion of IASI NH3 valid pixels was generally lower than GIIRS, most of which were around 35%, the spatial distribution of IASI NH3 effective pixels in China was relatively uniform without concentration. We had performed interpolation processing to ensure the spatial continuity and integrity of IASI NH3 column. In general, the number of available pixels from GIIRS and IASI met the analysis requirements.

We added the following sentences and figure in supplement.

---

## Author Comment (AC2)

**Comments:**

In this manuscript, the authors report on a study aimed at analyzing the changes of hourly $NH_3$ concentrations and estimating surface $NH_3$ concentrations and $NH_3$ emissions in China with top-down method. The manuscript fits into the scope of ACP and the results presented are very interesting to their readers. Overall the paper is clearly structured and generally well written. I have the following comments of the paper that should be addressed.

We thank the reviewer for your time and helpful comments. Our point-by-point response is enclosed.

**General comments:**

1. Although the sources of uncertainty in the experiments covered are described in the limitations and outlook section, a quantitative analysis is lacking and should be added. How did you solve the problem of missing GIIRS data in the Yangtze River Basin mentioned in the constraints?

We thank the reviewer for the comment. Surface $NH_3$ concentration is the key variable in $NH_3$ emission calculation. In this paper, the $NH_3$ measurements from the NNDMN in China were collected and compared with IASI-derived surface $NH_3$ concentrations. The regression $R_2$ between measured results and satellite-estimated annual means was 0.72 and the RMSE was 2.24 µg N m$^{-3}$. The coefficient of the fitted line was 1.03 ≈ 1, with the bias of 2.59%. The regression $R_2$ between monthly average IASI-derived $NH_3$ concentrations and measured $NH_3$ by month ranged from 0.38-0.84, and the RMSE ranged from 2.29-3.36 µg N m$^{-3}$, with the biases less than 30% for all months. Overall, the calculated annual and monthly average IASI-derived surface $NH_3$ concentrations showed good agreement with the

measurements of sites, and generally indicated the level of error in the surface $NH_3$ concentration estimates.

For the missing GIIRS $NH_3$ observations in the Yangtze River Basin, we used GIIRS $NH_3$ observations to analyze the regional daily variation of $NH_3$ concentrations in China, and to estimate monthly average surface $NH_3$ concentrations and study the spatial and temporal distribution. The absence of $NH_3$ column in the Yangtze River basin can be filled by spatial interpolation. We did not interpolate the GIIRS $NH_3$ column, as it weakly affected the analysis of the daily cycle of $NH_3$ concentrations in China. We had averaged the observations for the same period (5 hours interval), and the spatial missing values were greatly reduced. With the exception of the Yangtze River Basin, the distribution of $NH_3$ concentrations was relatively complete in other regions. In addition, the main missing fraction of monthly mean surface $NH_3$ concentrations from GIIRS was also found in a small part of the Tibetan Plateau, and interpolation was not carried out as it would introduce additional errors.

We added the following sentences into our manuscript.
"Third, the spatial resolution of the $NH_3$ vertical profile simulated by the atmospheric model is relatively coarse (0.5 degrees). In order to make it consistent with the spatial resolution of the remote sensing data, the outputs of GEOS-Chem (vertical profiles and feedback ratio between emissions and surface $NH_3$ concentrations) were interpolated through resampling methods. Owing to the resolution limit, the ratio-based mass balance approach to estimate $NH_3$ emissions neglected the effects of internal transport of $NH_3$ and displacement of emission sources within the fine grid.

Finally, there are some uncertainties and biases in the observed $NH_3$ column by satellite. Earlier versions of the IASI $NH_3$ column product were 25-50% lower than ground-based measurements (Whitburn et al., 2016; Dammers et al., 2017). However, the new version of IASI v3 lacks a comprehensive ground-based measurement assessment, which has only been compared with limited aircraft observations(Guo et al., 2021). Comparing IASI-derived surface $NH_3$ concentrations with measurements of ground sites (NNDMN) generally shows consistency in this study. The further work is needed for the complete assessment and error analysis."

2. I am confused about the treatment of the feedback ratio of surface $NH_3$ concentrations and emissions mentioned in the methodology. Is it the calculation done on an annual scale or on a monthly scale? Is it a variable value over time or a constant value? The feedback ratio should also be included as an element in the uncertainty and limitation analysis.

We thank the reviewer for the comment. We obtained the feedback ratio between surface $NH_3$ concentrations and $NH_3$ emissions using the mass balance method with GEOS-Chem simulation. In the study, the REAS emission inventory was used as China's anthropogenic emissions into GEOS-chem. However, as the time series range of REAS only corresponded to the IASI observations during 2008-2015, the feedback ratios for 2016-2019 were not obtained. Therefore, we used the fixed monthly average feedback ratio (Fig. S2b) for the calculation of $NH_3$ emissions.

We added the following figures in supplement.

[Figure]

**Figure S2.** Conversion ratios from GEOS-Chem simulations. (a) The conversion ratio of total $NH_3$ concentrations and surface $NH_3$ concentrations. (b) The feedback ratio of surface $NH_3$ concentrations and $NH_3$ emissions.

**Minor comments:**

line 247: check and modify the content in the Figure 5

Figure 5 showed satellite-derived surface $NH_3$ concentrations compared to ground-based measurements from 2008-2015. There were some errors in the description and we have revised them.

"Monthly regression $R^2$ between the satellite-derived $NH_3$ concentration and the measured $NH_3$ was 0.38-0.84. The regression $R^2$ reached the higher value (>0.80) in July and August. The RMSE ranged from 2.29- 3.36 $\mu g \ N \ m^{-3}$, which reached the maximum value of 3.36 $\mu g \ N \ m^{-3}$ in July, and reached the smallest in March (2.29 $\mu g \ N \ m^{-3}$). The bias is basically less than 31% for all months, and reached the minimum

value of -0.67% in February, indicating that the monthly IASI-derived surface concentration obtained are

consistent with measurements."

[Figure]

**Figure 5.** Comparison of monthly average values of IASI-derived and observed NH₃ surface

concentrations in 2010-2015.

line 292: The data of Figure. 8 doesn't match the data described in the article

We thank the reviewer for pointing this out. Figure 8 showed the yearly change in IASI-derived NH₃

emissions over China from 2008-2019, the monthly change in NH₃ emissions in 2019 and the spatial

distribution of $NH_3$ emissions in January, April, July and October 2019. There were some errors in the description and we have revised them.

"Based on the top-down estimates, China's $NH_3$ emissions ranged from 12.17-17.77 Tg N yr$^{-1}$ during 2008-2019. From 2008 to 2015, $NH_3$ emissions increased from 13.00 Tg N yr$^{-1}$ to 17.06 Tg N yr$^{-1}$. Since 2008, the temperature in China has risen steadily (Ding et al., 2007), which promotes the volatilization of $NH_3$, which partly explains the increase in $NH_3$ emissions from 2008 to 2015. After 2015, $NH_3$ emissions fluctuated and changed slightly (16.08-17.77 Tg N yr$^{-1}$). Compared with other studies, the change in $NH_3$ emissions from 2008 to 2015 is consistent with previous estimates, and the overall $NH_3$ emissions show an upward trend (Kang et al., 2016; Zhang et al., 2018; Ma, 2020; Fu et al., 2020; Zhang et al., 2021). Our estimates are on the rise as a whole, but the calculated values are generally lower than those by (Fu et al., 2020) (around 15 Tg N yr$^{-1}$), but larger than those by EDGAR and Kang et al. (2016)."

[Figure]

**Figure 8**. Annual changes of NH₃ emissions (a), monthly changes of NH₃ emissions in 2019 (b) and spatial distribution of NH₃ emissions by month in 2019 (c).

line 13: replace "China has largest NH$_3$ emissions in the world…" by "China has the largest NH$_3$ emissions globally…"

We have changed it as suggested. "China has the largest NH$_3$ emissions globally, mainly associated with agricultural sources including nitrogen fertilizer and livestock."

line 89: replace "method by using" with "method using"

We have changed "by using" to " using".

line 103: Correct "are" to be " is"

It is fixed now.

line 115: replace "at high frequency" with "at high frequencies"

We have changed it as suggested.

line 121: change "The average value of HRI is 0 with the standard deviation as 1" to be "The average value of HRI is 0 with a standard deviation of 1"

We have changed it as suggested. "The average value of HRI is 0 with a standard deviation of 1, and the HRI range is [-1,1]. "

line 124: replace "from November in 2019 to October in 2020" to be "from November 2019 to October 2020"

We have changed it as suggested. " In this study, we used hourly $NH_3$ concentrations during 2019-2020 (from November 2019 to October 2020) to study $NH_3$ diel cycle with a resolution of 0.5°."

line 139: change "product of" to be "product of the"

We have changed it as suggested.

line 168: Correct "which is" with "which are"

Corrected.

line 208: replace "while for other time…." with "while $NH_3$ concentration … at other times"

We have changed it as suggested. "…while $NH_3$ concentration tends to be stable at other times."

line 208: replace "changes of" to be "changes in"

We have changed it as suggested.

line 214: replace "which may be also related with" by "which may also be related to"

We have changed it as suggested.

line 215: replace "except" by "except for" and replace "have" by "has"

We have changed it as suggested.

line 216: Correct "patterns" by "patterns are"

Corrected.

line 253: delete "during 2010-2015"

We have removed the " during 2010-2015".

line 267: replace "have" by "had"

We have changed it as suggested.

line 276: replace "change of" to be "change in"

It is fixed now.

line 305: Correct "are" to be "is"

Corrected.

line 315: change "estimated" to be "estimate"

We have changed it as suggested.

line 318: change "in" to be "from"

Fixed.

line 320: change "low" to be "the low"

It is fixed now.

line 325: change "occurred" to be "occur"

We have changed it as suggested.

**Reference**

Dammers, E., Shephard, M. W., Palm, M., Cady-Pereira, K., Capps, S., Lutsch, E., Strong, K., Hannigan, J. W., Ortega, I., Toon, G. C., Stremme, W., Grutter, M., Jones, N., Smale, D., Siemons, J., Hrpcek, K., Tremblay, D., Schaap, M., Notholt, J., and Erisman, J. W.: Validation of the CrIS fast physical $NH_3$ retrieval with ground-based FTIR, Atmos. Meas. Tech., 10, 2645-2667, https://doi.org/10.5194/amt-10-2645-2017, 2017.

Ding, Y. H., Ren, G. Y., Zhao, Z. C., Xu, Y., Luo, Y., Li, Q. P., and Zhang, J.: Detection, causes and projection of climate change over China: An overview of recent progress, Adv. Atmos. Sci., 24, 954-971, https://doi.org/10.1007/s00376-007-0954-4, 2007.

Fu, H., Luo, Z., and Hu, S.: A temporal-spatial analysis and future trends of ammonia emissions in China, Sci. Total Environ., 731, 138897, https://doi.org/10.1016/j.scitotenv.2020.138897, 2020.

Guo, X., Wang, R., Pan, D., Zondlo, M. A., Clarisse, L., Van Damme, M., Whitburn, S., Coheur, P. F., Clerbaux, C., Franco, B., Golston, L. M., Wendt, L., Sun, K., Tao, L., Miller, D., Mikoviny, T., Müller, M., Wisthaler, A., Tevlin, A. G., Murphy, J. G., Nowak, J. B., Roscioli, J. R., Volkamer, R., Kille, N., Neuman, J. A., Eilerman, S. J., Crawford, J. H., Yacovitch, T. I., Barrick, J. D., and Scarino, A. J.: Validation of IASI satellite ammonia observations at the pixel scale using in situ vertical profiles, J Geophys Res-Atmos, 126, e2020JD033475, https://doi.org/10.1029/2020jd033475, 2021.

Kang, Y., Liu, M., Song, Y., Huang, X., Yao, H., Cai, X., Zhang, H., Kang, L., Liu, X., Yan, X., He, H., Zhang, Q., Shao, M., and Zhu, T.: High-resolution ammonia emissions inventories in China from 1980 to 2012, Atmos. Chem. Phys., 16, 2043-2058, https://doi.org/10.5194/acp-16-2043-2016, 2016.

Ma, S.: High-resolution assessment of ammonia emissions in China: Inventories, driving forces and mitigation, Atmos. Environ., 229, 117458, https://doi.org/10.1016/j.atmosenv.2020.117458, 2020.

Whitburn, S., Van Damme, M., Clarisse, L., Bauduin, S., Heald, C., Hadji-Lazaro, J., Hurtmans, D., Zondlo, M. A., Clerbaux, C., and Coheur, P. F.: A flexible and robust neural network IASI-$NH_3$ retrieval algorithm, J Geophys Res-Atmos, 121, 6581-6599, https://doi.org/10.1002/2016JD024828, 2016.

Zhang, L., Chen, Y. F., Zhao, Y. H., Henze, D. K., Zhu, L. Y., Song, Y., Paulot, F., Liu, X. J., Pan, Y. P., Lin, Y., and Huang, B. X.: Agricultural ammonia emissions in China: reconciling bottom-up and top-down estimates, Atmos. Chem. Phys., 18, 339-355, https://doi.org/10.5194/acp-18-339-2018, 2018.

Zhang, X. M., Ren, C. C., Gu, B. J., and Chen, D. L.: Uncertainty of nitrogen budget in China, Environ. Pollut., 286, 9, https://doi.org/10.1016/j.envpol.2021.117216, 2021.

---

## Author Comment (AC3)

**Comments:**

The study estimated the changes of hourly $NH_3$ concentrations, surface $NH_3$ concentrations and $NH_3$ emissions in China using the polar-orbiting satellite (IASI) and Fengyun-4 geostationary satellite. The results show $NH_3$ concentration in daytime was generally higher than that at night. Satellite-based $NH_3$ emissions ranged from 12.99-17.77 Tg N $yr^{-1}$ during 2008-2019. The manuscript is overall well organized and written. The analyses are neatly conducted and fit the scope of ACP. Before recommending publish the study, I have the following comments that I think the authors shall address to improve the manuscript.

We thank the reviewer for your time and helpful comments. Our point-by-point response is enclosed.

**General comments:**

1. Throughout the paper, there are issues with the use of plural vs singular and or verb tense, especially in the use of 3rd person, plural or singular. There is also extensive mixed use of past tense and present tense instead. I strongly recommend using unified tense instead, throughout the paper.

Thanks, we have re-checked the use of plural/singular, used the present tense, and conducted a careful copyediting throughout the paper.

2. The feedback between surface $NH_3$ concentration and emissions was calculated by GEOS-Chem. Please describe the simulation process in detail and driven data in SI.

We thank the reviewer for the comment. The study used the global 3-D chemistry transport model GEOS-Chem v12.3.0 , developed by Harvard University, to simulate and calculate the conversion ratio of surface $NH_3$ concentrations and emissions, which was widely used in the field of atmospheric physical chemistry

research (Chen et al., 2009; Zhang et al., 2011). GEOS-Chem contained detailed tropospheric gas-aerosol ($O_3$, $NO_x$, $NH_3$, $SO_4^{2-}$, $NO_3^-$ and $NH_4^+$, etc.) chemistry, and driven by assimilated meteorological information from the NASA Goddard Earth Observing System (GEOS-5) (https://gmao.gsfc.nasa.gov/). The source programs of GEOS-Chem were freely available from Atmospheric Chemistry Modeling Group at Harvard University (http://acmg.seas.harvard.edu/geos/geos_overview.html). The driver input files contained meteorological data and emission inventory, and the directory file contained the initial concentration file, photolysis mechanism and chemical mechanism files.

In this study, the nested regional model for Asia was driven by assimilated GEOS-5 meteorological data with a horizontal resolution of $1/2^{\circ} \times 2/3^{\circ}$. The GEOS-Chem model here did not consider land-atmosphere bi-directional $NH_3$ exchange, and the $NH_3$ flux was parameterized as uncoupled emission and dry deposition processes. Anthropogenic emissions in China are from the Regional Emission in Asia (REAS-v2) inventory. The GEOS-Chem $NH_3$ concentrations output includes 47 layers from the ground to the top of the atmosphere to capture the $NH_3$ vertical profiles. Only the inventory data overlapped with the IASI measured period were used, which was sampled and consistent with the model resolution. We have added the following figure in the supplement.

[Figure]

The ratio of total NH$_3$ column and surface concentrations   The ratio of surface NH$_3$ concentrations and emissions

**Figure S2.** Conversion ratios from GEOS-Chem simulations. (a) The conversion ratio of total NH$_3$ concentrations and surface NH$_3$ concentrations. (b) The feedback ratio of surface NH$_3$ concentrations and NH$_3$ emissions.

**Minor comments:**

L56, please further check if the value is 40%, if so, annual farmland NH$_3$ emission were estimated as 2.4 Tg N yr$^{-1}$ by the IPCC tier 1 guidelines.

We have further checked the citation and the method in the paper calculates NH$_3$ emissions from China's farmland to be 40% higher than the IPCC in 2008. However, we lacked clarification and the data used in the paper were for the specific year. We have made the following changes to the original text.

"Zhou et al. (2016) calculated the annual farmland $NH_3$ emission ($3.96 \pm 0.76$ Tg N $yr^{-1}$) over China in 2008 based on the bottom-up method, which is 40% higher than the emission in the Intergovernmental Panel on Climate Change (IPCC) Tier 1 guidelines (2.89 Tg N $yr^{-1}$)."

L29, 57, what's $SO_2$, NOx, $NH_4^+$, and IPCC, etc.

$NH_3$ reacts with acid pollutants (Sulfur dioxide ($SO_2$) and nitrogen oxides ($NO_x$)) to form fine particulate matters (such as PM2.5 (particles less than 2.5 micrometers in diameter)), leading to haze pollution. In addition, the deposition of $NH_3$ and ammonium ($NH_4^+$) could also cause environmental problems such as water eutrophication, biodiversity loss and soil acidification (Paerl et al., 2014).

We have added the following content to explain them. "…, which is 40% higher than the emission in the Intergovernmental Panel on Climate Change (IPCC) Tier 1 guidelines (2.89 Tg N $yr^{-1}$)."

L113, correct word 'is' to 'are', please check similar mistake throughout the manuscript.

It is fixed now, and we have double-checked and corrected similar errors in the manuscript

L201, in Figure 1, check $NH_3$ concentrations (kg N $ha^{-1}$) or $NH_3$ concentrations (ppb).

We have revised the units of $NH_3$ concentrations in Figure 1 to $10^{15}$ molecules $cm^{-2}$.

L201, in Figure 1, the figure shows the 2019-2020 average or sum, please check similar mistake throughout the manuscript.

The time series of NH$_3$ columns by GIIRS was between 2019.11 and 2020.10. We showed the spatial variation of the monthly average NH$_3$ columns at 10 overpass times for GIIRS from 2019-2020. We have revised the similar mistake throughout the paper and made the following changes to the figure captions.

[Figure]

**Figure 1.** Monthly average NH$_3$ columns for each of the 10 GIIRS overpass time periods during 2019-2020.

L282-283, need the data link or reference.

We have added references as suggested.

"The North China Plain is China's granary, with developed agriculture and animal husbandry, high population densities and strong human activities (including vehicle emissions) (Zhang et al., 2006; Wang et al., 2020)."

**Reference**

Chen, D., Wang, Y., McElroy, M. B., He, K., Yantosca, R. M., and Le Sager, P.: Regional CO pollution and export in China simulated by the high-resolution nested-grid GEOS-Chem model, Atmos. Chem. Phys., 9, 3825-3839, https://doi.org/10.5194/acp-9-3825-2009, 2009.

Wang, Z., Zhang, X., Liu, L., Cheng, M., and Xu, J.: Spatial and seasonal patterns of atmospheric nitrogen deposition in North China, Atmospheric and Oceanic Science Letters, 13, 188-194, https://doi.org/10.1080/16742834.2019.1701385, 2020.

Zhang, L., Jacob, D. J., Downey, N. V., Wood, D. A., Blewitt, D., Carouge, C. C., van Donkelaar, A., Jones, D. B., Murray, L. T., and Wang, Y.: Improved estimate of the policy-relevant background ozone in the United States using the GEOS-Chem global model with $1/2 \times 2/3$ horizontal resolution over North America, Atmos. Environ., 45, 6769-6776, https://doi.org/10.1016/j.atmosenv.2011.07.054, 2011.

Zhang, Y., Liu, X., Zhang, F., Ju, X., Zou, G., and Hu, K.: Spatial and temporal variation of atmospheric nitrogen deposition in the North China Plain, Acta Ecologica Sinica, 26, 1633-1638, https://doi.org/10.1016/S1872-2032(06)60026-7, 2006.

Zhou, F., Ciais, P., Hayashi, K., Galloway, J., Kim, D.-G., Yang, C., Li, S., Liu, B., Shang, Z., and Gao, S.: Re-estimating $NH_3$ emissions from Chinese cropland by a new nonlinear model, Environ. Sci. Technol., 50, 564-572, https://doi.org/10.1021/acs.est.5b03156, 2016.

---

## Author Response (AR1)

**Response to Reviewer #1:**

**Comments:**

In this manuscript, the authors report on a study aimed at analyzing the changes of hourly NH3 concentrations and estimating surface NH3 concentrations and NH3 emissions in China with top-down method. The manuscript fits into the scope of ACP and the results presented are very interesting to their readers. Overall the paper is clearly structured and generally well written. I have the following comments of the paper that should be addressed.

**General comments:**

1. Although the sources of uncertainty in the experiments covered are described in the limitations and outlook section, a quantitative analysis is lacking and should be added. How did you solve the problem of missing GIIRS data in the Yangtze River Basin mentioned in the constraints?

Surface NH3 concentration is the key variable in NH3 emission calculation. In this paper, the NH3 measurements from the NNDMN in China were collected and compared with IASI-derived surface NH3 concentrations. The regression R2 between measured results and satellite-estimated annual means was 0.72 and the RMSE was 2.24  $\mu$ g N m-3. The coefficient of the fitted line was 1.03  $\approx$  1, with the bias of 2.59%. The regression R2 between monthly average IASI-derived NH3 concentrations and measured NH3 by month ranged from 0.38-0.84, and the RMSE ranged from 2.29-3.36  $\mu$ g N m-3, with the biases less than 30% for all months. Overall, the calculated annual and monthly average IASI-derived surface NH3 concentrations showed good agreement with the measurements of sites, and generally indicated the level of error in the surface NH3 concentration estimates.

For the missing GIIRS NH3 observations in the Yangtze River Basin, we used GIIRS NH3 observations to analyze the regional daily variation of NH3 concentrations in China, and to estimate monthly average surface NH3 concentrations and study the spatial and temporal distribution. The absence of NH3 column in the Yangtze River basin can be filled by spatial interpolation. We did not interpolate the GIIRS NH3 column, as it weakly affected the analysis of the daily cycle of NH3 concentrations in China. We had averaged the observations for the same period (5-hour interval), and the spatial missing values were greatly reduced. With the exception of the Yangtze River Basin, the distribution of NH3 concentrations was relatively complete in other regions. In addition, the main missing fraction of monthly mean surface NH3 concentrations from GIIRS was also found in a small part of the Tibetan Plateau, and interpolation was not carried out as it would introduce additional errors.

We added the following sentences into our manuscript.

"Third, the spatial resolution of the NH3 vertical profile simulated by the atmospheric model is relatively coarse (0.5 degrees). In order to make it consistent with the spatial resolution of the remote sensing data, the outputs of GEOS-Chem (vertical profiles and feedback ratio between emissions and surface NH3 concentrations) were interpolated through resampling methods. Owing to the resolution limit, the ratiobased mass balance approach to estimate NH3 emissions neglected the effects of internal transport of NH3 and displacement of emission sources within the fine grid.

Finally, there are some uncertainties and biases in the observed NH3 column by satellite. Earlier versions of the IASI NH3 column product were 25-50% lower than ground-based measurements (Whitburn et al.,

2016; Dammers et al., 2017). However, the new version of IASI v3 lacks a comprehensive ground-based measurement assessment, which has only been compared with limited aircraft observations(Guo et al., 2021). Comparing IASI-derived surface NH3 concentrations with measurements of ground sites (NNDMN) generally shows consistency in this study. The further work is needed for the complete assessment and error analysis."

2. I am confused about the treatment of the feedback ratio of surface  $NH_3$  concentrations and emissions mentioned in the methodology. Is it the calculation done on an annual scale or on a monthly scale? Is it a variable value over time or a constant value? The feedback ratio should also be included as an element in the uncertainty and limitation analysis.

We obtained the feedback ratio between surface NH3 concentrations and NH3 emissions using the mass balance method with GEOS-Chem simulation. In the study, the REAS emission inventory was used as China's anthropogenic emissions into GEOS-chem. However, as the time series range of REAS only corresponded to the IASI observations during 2008-2015, the feedback ratios for 2016-2019 were not obtained. Therefore, we used the fixed monthly average feedback ratio (Fig. S2b) for the calculation of NH3 emissions.

We added the following figures in the supplement.

**Figure S2.** Conversion ratios from GEOS-Chem simulations. (a) The conversion ratio of total NH3 concentrations and surface NH3 concentrations. (b) The feedback ratio of surface NH3 concentrations and NH3 emissions.

**Minor comments:**

line 247: check and modify the content in the Figure 5

Figure 5 showed satellite-derived surface NH3 concentrations compared to ground-based measurements from 2008-2015. There were some errors in the description and we have revised them.

"Monthly regression  $R^2$  between the satellite-derived NH3 concentration and the measured NH3 was 0.38-0.84. The regression  $R^2$  reached the higher value (>0.80) in July and August. The RMSE ranged from 2.29- 3.36 µg N m-3, which reached the maximum value of 3.36 µg N m-3 in July, and reached the smallest in March (2.29 µg N m-3). The bias is basically less than 31% for all months, and reached the minimum value of 0.67% in February, indicating that the monthly IASI-derived surface concentration obtained are